# The Link between Endogenous Pain Modulation Changes and Clinical Improvement in Fibromyalgia Syndrome: A Meta-Regression Analysis

**DOI:** 10.3390/biomedicines12092097

**Published:** 2024-09-13

**Authors:** Kevin Pacheco-Barrios, Rafaela Machado Filardi, Luis Fernando González-González, Nayeon Park, Fernanda Queiroz Petrus, Alba Navarro-Flores, Silvia Di-Bonaventura, Luana Gola Alves, Fernanda Queiroz, Felipe Fregni

**Affiliations:** 1Neuromodulation Center and Center for Clinical Research Learning, Spaulding Rehabilitation Hospital and Massachusetts General Hospital, Harvard Medical School, Boston, MA 02115, USA; rfilardi@mgb.org (R.M.F.); lgonzalez-gonzalez@mgb.org (L.F.G.-G.); parknayeon021@gmail.com (N.P.); fqueirozpetrus@gmail.com (F.Q.P.); sdibonaventura@mgb.org (S.D.-B.); luanagola@gmail.com (L.G.A.); fdequeirozsilva@mgb.org (F.Q.); 2Unidad de Investigación para la Generación y Síntesis de Evidencias en Salud, Vicerrectorado de Investigación, Universidad San Ignacio de Loyola, Lima 15023, Peru; 3Admission AG, Irvine, CA 92618, USA; 4Escuela de Medicina, Universidad Cesar Vallejo, Trujillo 13001, Peru; alba0736@gmail.com; 5Department of Physical Therapy, Occupational Therapy, Rehabilitation and Physical Medicine, Rey Juan Carlos University, 28922 Alcorcon, Spain; 6Cognitive Neuroscience, Pain and Rehabilitation Research Group (NECODOR), Faculty of Health Sciences, Rey Juan Carlos University, 28922 Alcorcon, Spain

**Keywords:** conditioned pain modulation, temporal summation, fibromyalgia, chronic pain, biomarker

## Abstract

Conditioned pain modulation (CPM) and temporal summation (TS) tests can measure the ability to inhibit pain in fibromyalgia syndrome (FMS) patients and its level of pain sensitization, respectively. However, their clinical validity is still unclear. We studied the association between changes in the CPM and TS tests and the clinical improvement of FMS patients who received therapeutic intervention. We systematically searched for FMS randomized clinical trials with data on therapeutic interventions comparing clinical improvement (pain intensity and symptom severity reduction), CPM, and TS changes relative to control interventions. To study the relationship between TS/CPM and clinical measures, we performed a meta-regression analysis to calculate odds ratios. We included nine studies (484 participants). We found no significant changes in TS or CPM by studying all the interventions together. Our findings show that this lack of difference is likely because pharmacological and non-pharmacological interventions resulted in contrary effects. Non-pharmacological interventions, such as non-invasive neuromodulation, showed the largest effects normalizing CPM/TS. Meta-regression was significantly associated with pain reduction and symptom severity improvement with normalization of TS and CPM. We demonstrate an association between clinical improvement and TS/CPM normalization in FMS patients. Thus, the TS and CPM tests could be surrogate biomarkers in FMS management. Recovering defective endogenous pain modulation mechanisms by targeted non-pharmacological interventions may help establish long-term clinical recovery in FMS patients.

## 1. Introduction

Fibromyalgia syndrome (FMS) is a chronic rheumatic disease characterized by diverse symptoms such as widespread body pain, stiffness, fatigue, sleep disturbances, cognitive impairments, and psychiatric symptoms, which significantly impact the patient’s daily routine and quality of life. Furthermore, it often requires healthcare services, becoming a costly public health issue [1].

Central sensitization consists of a variety of neural system dysfunctions, and it provides a possible explanation for comprehension and management of chronic pain syndromes such as fibromyalgia [2]. This theory suggests that abnormal sensory processing in the brain, increased susceptibility to stimulus, and inaccurate pain modulation pathways result in hypersensitivity to stimulus [2,3]. In addition, Mezhov et al. (2021) [3] argue that the presence of central sensitivity syndromes, such as fibromyalgia, may reflect an overactive central nervous system instead of an isolated disorder, associated with changes in brain oscillatory activity [4,5]. Recognizing central sensitization as a mechanism in chronic pain increases the importance of treatments focusing on modulating the endogenous pain inhibitory system [6] instead of short-term treatment that only provides temporary pain relief by blocking nociceptive input. This change in pain management provides awareness and highlights the importance of targeting pain compensatory mechanisms as a new approach to manage FMS [2].

Conditioned pain modulation (CPM) and pain temporal summation (TS) are key to understanding pain modulation dynamics and can assess the pain sensitization and pain inhibition characteristics of FMS patients non-invasively; they are also promising biomarkers [7]. CPM involves the inhibition of one painful stimulus by another, reflecting the body’s endogenous pain control systems and assessing their efficiency in both healthy and chronic-pain populations [7]. TS describes the increased perception of pain from repeated noxious stimuli, highlighting excitatory processes in central pain facilitation [8]. However, there is limited evidence supporting the validity of these biomarkers, mainly due to contradictory results associating CPM metrics with clinical outcomes. O’Brien et al. (2018) found that FM patients show a heightened response to repetitive noxious stimuli and reduced CPM compared to healthy controls, supporting the idea of central sensitization and defective pain inhibition [2]. Similarly, Potvin and Marchand (2016) reported that CPM procedure has a specificity of 78.9%, but its low sensitivity of 45.7% indicates that it may not be detected in all patients [9]. These findings suggest that endogenous pain inhibition mechanisms are impaired in these patients, but only partially, which indicates the complexity and unpredictability of pain modulation in this condition. Conversely, a study found no correlation between pain severity and CPM efficacy, challenging the idea that CPM is a reliable biomarker [10]. Similarly, a different study reported no significant difference in CPM between FM patients and those with generalized pain, indicating diverse pain phenotypes and mechanisms in FM syndrome [11]. The limited sample size and the cross-sectional nature of the analysis, not considering the dynamic features of endogenous pain modulation, indicate the presence of limitations in previous studies. To our knowledge, no previous studies tried to validate systematically the longitudinal changes in CPM/TS with changes in clinical outcomes such as pain intensity or symptoms severity in FMS. A longitudinal validation of these biomarkers is needed to foster the utilization of objective metrics in the context of FMS management and the development of personalized therapeutic protocols.

Therefore, we aim to explore the association between changes of endogenous pain modulation (indexed by CPM and TS) and clinical changes (pain intensity and symptoms severity) after therapeutic interventions in FMS patients. We hypothesize that improvement in endogenous pain modulation will correlate with the improvement of clinical profiles in FMS patients.

## 2. Materials and Methods

This systematic review and meta-analysis were conducted following the “Preferred Reporting Items for Systematic Reviews and Meta-Analyzes” (PRISMA) [12] guidelines and the Cochrane Handbook for Systematic Reviews of Interventions [13]. The protocol was registered from the Open Science Framework platform (https://osf.io/registries (accessed on 9 August 2024)) with the code YDG4J (DOI: https://doi.org/10.17605/OSF.IO/YDG4J).

### 2.1. Search Strategy and Inclusion Criteria

PubMed/MEDLINE and Embase databases were searched from inception until 4 March 2024. The complete search strategy was: (“Fibromyalgia” [MeSH] OR “Fibromyalgia” [TIAB]) AND (“Sensory profile” OR “temporal summation” OR “conditioned pain modulation” OR “temporal slow pain summation” OR “quantitative sensory testing” OR “cognitive-emotional sensitization” OR “sensory threshold” OR “pain threshold” OR “diffuse noxious inhibitory control” OR “heterotopic noxious conditioning stimulation” OR “endogenous analgesia” OR “pain inhibition” OR “endogenous pain modulation”). Additionally, we reviewed the references of the included studies. The eligibility criteria were: (1) Randomized controlled trials (RCTs) that included FM patients and assessed CPM/TS before and after an intervention and that included at least one clinical pain measure (e.g., VAS, FIQR); (2) any type of intervention and comparison; and (3) full-text accessible. The exclusion criteria were: (1) pre-clinical studies; (2) review articles; (3) letters to the editor and editorials; and (4) conference abstracts. Studies were not excluded based on date or language.

### 2.2. Study Selection and Data Extraction

Duplicate records were removed manually using the Covidence web platform. Two independent reviewers conducted the screening, first by titles and abstracts and then by full text, with the aid of the Covidence web platform. Disagreements were solved by the facilitation of a third author. The data extraction was conducted by two authors independently. The following variables of interest were extracted: study design, sample size, CPM/TS method, type of stimulus, type of control group, intervention type, fibromyalgia diagnosis description, age, gender, pain intensity, and symptom severity scores indexed by the Revised Fibromyalgia Impact Questionnaire (FIQR).

### 2.3. Management of Missing Data

Unavailable raw data from the main outcome were calculated from available graphs using Web Plot Digitizer v.3.11 when possible. In the cases where data from graphs were not available, authors were contacted via email. If no response was received by the time of the analysis, the study was excluded.

### 2.4. Risk of Bias Assessment

Assessment of the studies’ risk of bias was carried by two independent reviewers, and discrepancies were solved with the support of a third author. We used the Cochrane Risk of Bias 2 (RoB 2) tool [14]. We used the algorithm proposed in the RoB 2 tool to rate each domain as ‘low risk of bias’, ‘some concerns’, or ‘high risk of bias’, with the overall risk of bias being the worst assessment of the five domains [15]. We did not conduct a publication bias assessment due to the small number of included studies.

### 2.5. Statistical Analysis

We conducted meta-analyses of continuous outcomes using random-effects models due to the high expected heterogeneity. The DerSimonian–Laird method was used [16]. Each variable (visual analogue scale (VAS), FIQR, CPM, and TS) was analyzed separately. Additionally, we converted the CPM and TS differences as percentage change as sensitivity analysis. Then, we calculated effect sizes as standardized mean differences (MD) with 95% confidence intervals (CIs) [13].

Moreover, we used the Hartung–Knapp adjustment for random effects models, which calculates more adequate error rates, especially when the number of included studies is small [17]. We tested for statistical heterogeneity of pooled estimates using the Chi^2^ test and the degree of heterogeneity using the *I*^2^ statistic, considering that heterogeneity might not be substantial when *I*^2^ < 40% [18]. We did not perform a publication bias assessment due to the low number of studies we found [19].

For the meta-regression analyses, we categorized each improvement in CPM and TS as an increase of at least 10% from the baseline in each trial arm (active and control). Similarly, we categorized the studies as clinical improvement based on an improvement on at least 20% of the scale from baseline, based on the minimal clinically important difference (MCID) for the FIQR score and the VAS score [20]. Then, we constructed two-by-two tables to calculate individual odds ratios (OR) and corresponding 95% CI using the Woolf approximation [21]. The data were processed with R Studio 4.1.1 for the creation of the plots (R Foundation for Statistical Computing, Vienna, Austria).

## 3. Results

Our literature search retrieved 1598 records. Of those, nine were included after full-text assessment (484 participants) [22,23,24,25,26,27,28,29,30]. The flowchart of the selection process is presented in Figure 1.

The interventions used were neuromodulation (n = 3), pharmacological (n = 5), and education (n = 1). Among these nine studies, four exclusively utilized CPM, one used only TS, and four incorporated both paradigms. Mechanical modalities (pain pressure threshold (PPT)) emerged as the predominant choice for conditioned stimulation (n = 7), while heat sensors were utilized in the remaining studies (n = 2). Concerning the conditioning stimulation in CPM, the cold water was the most commonly used (n = 4), with occlusion cuffs (n = 3) and hot water (n = 1) also employed in other studies. A detailed description of the included studies is found in Table 1.

### 3.1. Risk of Bias Findings

Overall, the studies demonstrated appropriate randomization procedures and adhered closely to the intended interventions, contributing to an overall assessment of low bias risk. However, one study presented some concerns regarding missing outcome data, primarily due to a significant number of participants in the intervention group experiencing collateral effects, which resulted in a high withdrawal rate from the study [26]. Another study was judged to have some concerns regarding deviations from intended interventions in certain studies, primarily due to the absence of blinding and the lack of a sham intervention [29]. These factors potentially introduced biases, as they could influence the result of the outcome. The full assessment is presented in Table 2.

### 3.2. Pooled Analysis Findings

Based on the data from the included studies, VAS scores were analyzed to compare the effects of the experimental intervention against the control group, showing a decrease in VAS scores in intervention group. However, the pooled effect size indicated a reduction in VAS scores with a wide confidence interval that suggests the overall effect is not statistically significant, SMD = 1.15, 95% CI [−3.70; 1.40]. The heterogeneity test revealed significant variability (I^2^ = 95%, *p* < 0.01), meaning that the intervention effects were inconsistent across different studies (Figure 2A).

The analysis of TS demonstrated varied effects between the experimental and control groups. The pooled effect size SMD = 0.48, 95% CI [−2.96; 3.92], indicated an increase in TS scores with a wide confidence interval not significant. The test of heterogeneity also revealed significant variability among the studies (I^2^ = 96%, *p* < 0.01), implying the intervention effects were inconsistent across different studies (Figure 2B).

After TS was analyzed using percentage transformations, the results showed varied effects between the experimental and control groups. The pooled effect size indicated a decrease in TS percentages. However, the overall effect was not statistically significant, SMD = −0.55, 95% CI [−3.80; 2.70]. The test of heterogeneity revealed high variability among the studies (I^2^ = 95%, *p* < 0.01) (Figure 2C).

Five out of nine studies reported FIQR scores, but the analyses demonstrated varied effects between the experimental and control groups. The pooled effect size was not statistically significant, SMD = −1.36, 95% CI [−7.25; 4.52]. The heterogeneity test showed high variability (*I*^2^ = 98%, *p* < 0.01) (Figure 3A). CPM raw values demonstrated a varied effect, SMD = −1.24, 95% CI [−10.94; 8.47], indicating a decrease in CPM, although the overall effect was not statistically significant. The weights of the studies are relatively evenly distributed. The test of heterogeneity revealed significant variability among the studies (I^2^ = 98%, *p* < 0.01). The non-significant overall effect combined with high heterogeneity might be attributable to the different types of interventions among studies and the various ways to measure CPM.

CPM values were analyzed using percentage transformations to compare the effects of the experimental intervention versus the control. The results demonstrated a varied effect on CPM percentages, SMD = 6.30, 95% CI [−0.50; 13.10], showing an increase in CPM percentages, although not statistically significant. The test of heterogeneity showed a high heterogeneity among the studies (*I*^2^ = 96%, *p* < 0.01). The meta-analysis explored the effects of various interventions on VAS scores, FIQR scores, TS, and CPM values (both raw and percentage-transformed for the last two). There is high variability among trials, with none of them having a significant overall effect. We can see that the both the VAS and FIQR scores showed a reduction in fibromyalgia due to the interventions. However, the overall effects remain non-significant (Figure 3B,C). Publication bias was not evaluated due to the small number of included studies.

### 3.3. Meta-Regression Findings

The meta-regression analysis revealed a significant association between clinical improvement and the normalization of CPM and TS. Patients included in studies that showed at least a 10% improvement in CPM and TS from baseline had higher odds of experiencing clinical improvement, defined as a minimum 20% reduction in FIQR and VAS scores. The calculated odds ratios (OR) indicated that the likelihood of clinical improvement was significantly greater in patients with improved CPM and TS (Table 3). These results underscore the potential of CPM and TS as biomarkers for treatment efficacy in fibromyalgia syndrome.

## 4. Discussion

Our meta-regression analysis examined the link between changes in CPM and TS tests and clinical improvement of pain in FMS patients. We found no significant overall changes in CPM or TS when all interventions were analyzed together. However, when differentiating between pharmacological and non-pharmacological interventions, we observed contrary effects. Our findings suggest that non-invasive neuromodulatory techniques, such as transcranial direct current stimulation (tDCS) and transcutaneous electrical nerve stimulation (TENS), may have considerable potential to normalize the CPM and TS metrics. Importantly, our analysis indicates a possible association between clinical improvement, specifically pain reduction and symptom severity, and the normalization of CPM and TS. These findings suggest that CPM and TS tests could be used in specific scenarios to guide and assess the efficacy of FMS treatments.

CPM and TS are essential in understanding the mechanisms of pain modulation. CPM has been established as a clinical tool in the assessment and management of chronic pain, providing valuable information on the ability of the central nervous system to regulate pain through endogenous mechanisms [31,32]. In patients with chronic pain, dysfunction of these inhibitory mechanisms is frequently observed [33,34], which also seems to be associated with an alteration in its CPM [35]. TS, which measures the facilitation of pain pathways, can indicate central sensitization, a key feature in conditions like FMS. Assessing CPM and TS in this specific population allows the identification of specific dysfunctions in pain modulation, facilitating a deeper understanding of the pathophysiological mechanisms underlying their condition.

One of the significant challenges in utilizing CPM and TS as biomarkers is the high variability observed across different paradigms used to assess these variables. This variability stems from differences in methodologies, including the type and intensity of conditioning and test stimuli, the anatomical sites tested, and the timing and duration of stimuli application. Such variability can lead to inconsistent results, making it difficult to compare findings across studies and limiting the generalizability of results [36]. The variability in CPM and TS paradigms presents both challenges and opportunities for research and clinical practice. On one hand, it highlights the need for standardized protocols to improve the reliability and reproducibility of assessments. Developing consensus guidelines for the test could enhance the consistency of research findings and facilitate more accurate comparisons across studies [37]. On the other hand, the variability also underscores the importance of tailoring CPM and TS assessments to individual patient needs. Different paradigms may reveal unique aspects of a patient’s pain modulation capacity, offering insights into specific dysfunctions in their endogenous pain control system. By understanding these nuances, clinicians can better personalize treatment strategies to address the distinct pain modulation profiles of their patients [36].

One relevant implication of measuring CPM is its ability to predict response to treatment [38,39,40]. Also, different studies have shown that the measurement of CPM can be useful to predict the response to conventional analgesic treatments, such as opioids, in different populations [41,42]. This predictive ability of CPM could allow clinicians to anticipate the effectiveness of therapeutic strategies and adjust them to maximize clinical benefits. For example, patients with preserved CPM function are more likely to respond positively to certain analgesics, providing a basis for personalized medicine in pain management.

Personalizing treatment based on pain modulation profile is another critical implication. By knowing a patient’s CPM and TS profiles, it may be possible to design a specific therapeutic regimen that directly addresses his or her individual needs [43]. Measurement of CPM and TS also plays an important role in monitoring treatment effectiveness over time [44]. This monitoring can allow clinicians to make therapeutic adjustments in near real time, ensuring optimal pain management. This approach aligns with the broader trend toward personalized medicine, which seeks to optimize treatment outcomes by considering individual variability in disease mechanisms and treatment responses.

Finally, CPM and TS can be used to classify patients into different phenotypes of chronic pain, allowing for a more accurate and effective therapeutic approach [45]. Future research should focus on establishing consistent methodologies for measuring CPM and TS, improving the robustness of these emerging biomarkers. Additionally, longitudinal studies are needed to explore the long-term effects of modulating CPM and TS and their impact on chronic pain progression and management.

### Strengths and Limitations

Our study has several strengths. Firstly, this is the first study exploring this topic in FMS patients. The comprehensive and systematic search strategy employed ensures a thorough examination of the available literature, enhancing the reliability of the findings. By including randomized clinical trials, the analysis benefits from high-quality data, providing robust evidence on the association between changes in CPM, TS, and clinical improvement in FMS patients. The use of meta-regression analysis allows for a nuanced understanding of how various therapeutic interventions influence CPM and TS, offering valuable insights into the underlying mechanisms of pain modulation in FMS. Moreover, the inclusion of both pharmacological and non-pharmacological interventions highlights the differential effects of these treatments, which is crucial for developing tailored therapeutic strategies.

However, there are limitations to consider. The heterogeneity of the included studies, in terms of interventions, outcome measures, and patient populations, may introduce variability that could affect the generalizability of the results. The relatively small number of studies and participants included in the analysis might limit the statistical power, assessment of publication bias, and precision of the meta-regression findings. Additionally, the reliance on published data means that potential unpublished studies with null results might be missing, leading to an overestimation of the effects. Despite these limitations, the review provides a valuable contribution to understanding the relationship between endogenous pain modulation and clinical outcomes in FMS.

## 5. Conclusions

In summary, this meta-regression suggests no significant changes in TS or CPM by studying all the interventions together. A significant association is revealed between clinical improvement (pain intensity and symptoms severity) and the normalization of CPM and TS in FMS patients. Despite the heterogeneity and limited number of included studies, our findings suggest that both pharmacological and non-pharmacological interventions can influence pain modulation mechanisms, improving the clinical pain measures CPM and TS. The CPM and TS tests may serve as valuable surrogate biomarkers for FMS management, highlighting the potential for targeted interventions to restore defective endogenous pain modulation mechanisms. Further research with larger sample sizes and standardized methodologies is needed to confirm these results and enhance clinical practice.

## Figures and Tables

**Figure 1 biomedicines-12-02097-f001:**
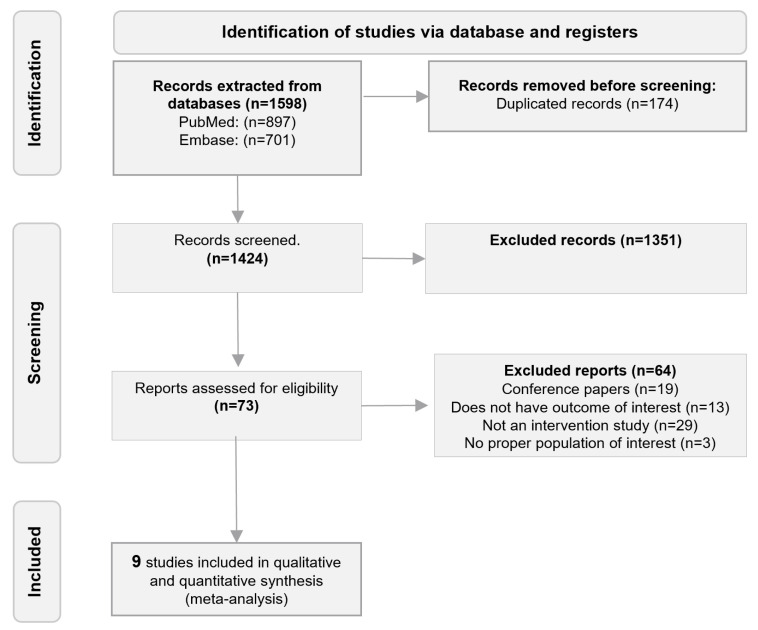
Flowchart of the study selection process.

**Figure 2 biomedicines-12-02097-f002:**
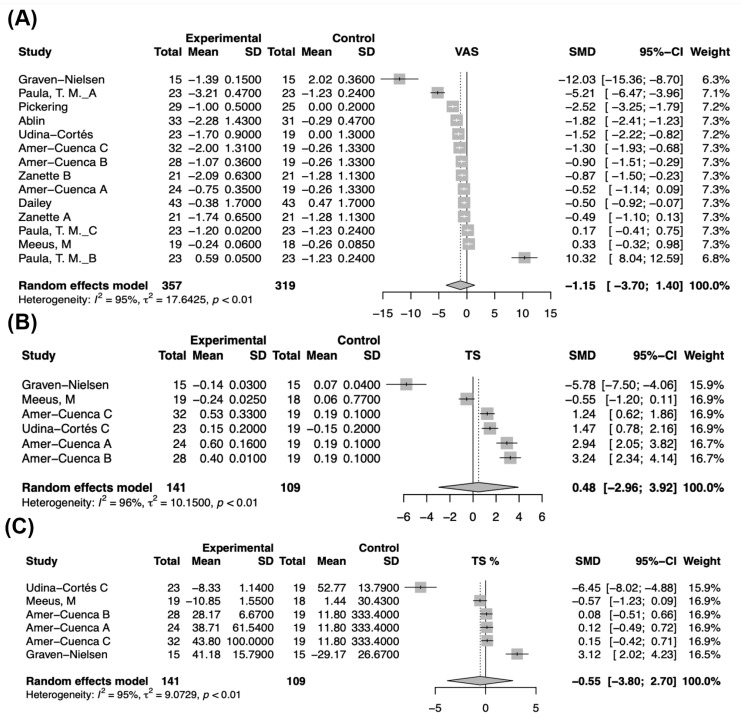
Forest plots of the meta-analysis of the main outcomes (VAS pain and TS) [22,23,24,25,26,27,28,29,30]. (**A**) VAS score changes, (**B**) Temporal summation raw score changes, (**C**) Temporal summation as percentage change.

**Figure 3 biomedicines-12-02097-f003:**
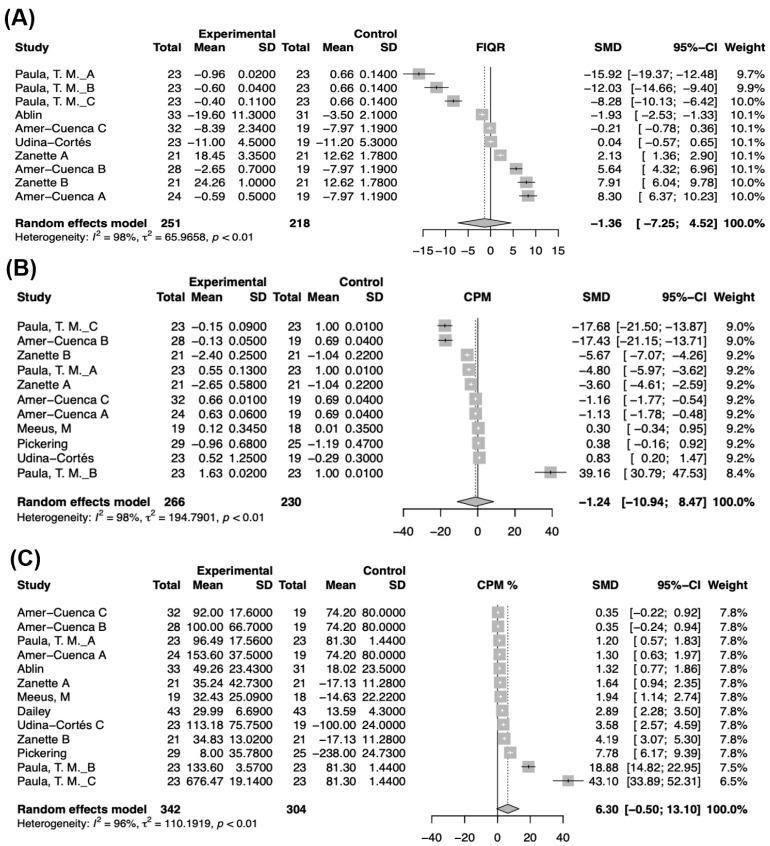
Forest plots of the meta-analysis of the main outcomes (FIQR pain and CPM) [23,24,25,26,27,28,29,30]. (**A**) FIQR score changes. (**B**) CPM raw score changes. (**C**) CPM as percentage change.

**Table 1 biomedicines-12-02097-t001:** General characteristics of included studies.

Study	Study Design	Age	Sample Size	Group Description	Type of Neuromodulator/Drug	Type of Conditioned Stimulation	Type of Conditioning Stimulation	CPM Method	TS Method	Measure of Clinical Pain Outcome
Graven-Nielsen et al., 2000 [22]	RCT CO	45 (8.25)	15	Group 1: Ketamine; Group 2: Placebo	Ketamine 0.3 mg/kg infussion per session	PPT (three bilateral tenderpoints)	NA	NA	PPT with 10 repeated applications of pulse. TS is the MD of VAS on the last pulse and the first pulse.	VAS
Dailey et al., 2013 [23]	RCT CO	49.1 (12.9)	43	Group 1: TENS; Group 2: Placebo	Transcutaneous electrical nerve stimulations (TENS), 100 Hz, 200 μs at maximal tolerable intensity.	PPT (cervical)	Cold Water	VAS assessed via PPT, realized with and without CS. CPM is the percentage of change of VAS between the two tests.	NA	VAS
Zanette et al., 2014 [24]	RCT	48.9 (8)	Group 1: 21 Group 2: 21 Group 3: 21	Group 1: Melatonin; Group 2: Amitriptyline + Melatonin; Group 3: Amitriptyline	Melatonin (10 mg) tablets + placebo; amitriptyline (25 mg) + placebo; or amitriptyline (25 mg) + melatonin (10 mg) once a day over 6-week period.	Heat pain (forearm)	Cold Water	VAS assessed via heat stimulation, measured with and without CS. CPM is the MD between the two tests.	NA	VAS FIQ
Meeus et al., 2015 [25]	RCT CO	44.58 (7.34)	19	Group 1: Acetaminophen; Group 2: Placebo	Acetaminophen 1 g—30 min before assessment	PPT (middle finger of right hand and trapezius right arm)	Occlusion cuff	NPRS assessed via PPT, realized with and without CS. CPM is the MD of NPRS between the two tests.	PPT with 10 repeated applications of pulse. TS is the MD of NPRS on the last pulse and the first pulse.	NPRS
Pickering et al., 2018 [26]	RCT	46.7 (10.6)	Group 1: 29 Group 2: 25	Group 1: Milnacipran; Group 2: Placebo	Milnacipran, titrated 50 mg days 1–3, 75 mg days 4–6, 100 mg days 7–1 months.	Heat pain (volar side of forearm)	Hot Water	NPRS assessed via PPT, realized with and without CS. CPM is the MD of NPRS between the two tests.	PPT with 10 repeated applications of pulse. TS is the MD of NPRS on the last pulse and the first pulse.	NPRS
Amer-Cuenca et al., 2019 [27]	RCT	53.4 (9.08)	Group 1: 24 Group 2: 28 Group 3: 32 Group 4: 19	Group 1: HD PNE; Group 2: LCD PNE; Group 3: LDD PNE; Group 4: Placebo	HP PNE (six 45 min sessions), LCD PNE (two 45 min sessions), LDD PNE (six 15 min sessions)	PPT (dorsal aspect of the distal phalanx)	Occlusion cuff	NPRS assessed via PPT, realized with and without CS. CPM is the MD of NPRS between the two tests.	PPT with 10 repeated applications of pulse. TS is the MD of NPRS on the last pulse and the first pulse.	NPRS FIQ
Udina-Cortes et al., 2020 [28]	RCT	52 (8)	Group 1: 20 Group 2: 17	Group 1: NAE; Group 2: NAE Sham	Self-Controlled Energo Neuro-Adaptive Regulator, 15–350 Hz, 4 ± 2 to 500 ± 50 milliseconds, 1.7–2.5 V to 100–150 V amplitude.	PPT (trapezius muscle)	Occlusion cuff	VAS assessed via PPT, realized with and without CS. CPM is the MD of VAS between the two tests.	PPT with 10 repeated applications of pulse. TS is the MD of VAS on the last pulse and the first pulse.	VAS FIQ
Ablin et al., 2023 [29]	RCT	45.04 (11.9)	Group 1: 33 Group 2: 31	Group 1: HBOT; Group 2: Duloxetina + Pregabalin	HBOT 60 daily sessions five times per week for a total of 12 weeks. 100% oxygen by mask at 2 absolute atmospheres (ATA) for 90 min.	PPT (upper trapezius muscle)	Cold Water	VAS assessed via PPT, realized with and without CS. CPM is the MD of VAS between the two tests.	NA	VAS FQI
Paula et al., 2023 [30]	RCT	49.3 (1.96)	Group 1: 21 Group 2: 22 Group 3: 22 Group 4: 21	Group 1: LDN + tDCS; Group 2: LDN + tDCS Sham; Group 3: Placebo + tDCS; Group 4: Placebo + tDCS Sham	tDCS using a current of 2 mA for 20 min/session (5 sessions). LDN 4.5 mg daily, orally for 26 days.	PPT (right forearm)	Cold Water	VAS assessed via PPT, realized with and without CS. CPM is the MD of VAS between the two tests.	NA	VAS FIQ

CO = crossover; CPM = conditioned pain modulation; CS = conditioning stimulation; FIQ = fibromyalgia impact questionnaire; HBOT = hyperbaric oxygen therapy; HD PNE: high-dose pain neuroscience education; LCD PNE = low-concentrated dose pain neuroscience education; LDD PNE low-diluted-dose pain neuroscience education; LDN = low-dose naltrexone; MD = mean difference; NA = not available; NAE = non-invasive neuro-adaptive electrostimulation; NPRS = numerical pain rating scale; PPT = pain pressure threshold; RCT = randomized controlled trial; tDCS = transcranial direct current stimulation; TENS = transcutaneous electrical nerve stimulation; TS = temporal summation; VAS = visual analogue scale. Values of age are given as a mean (SD).

**Table 2 biomedicines-12-02097-t002:** Risk of bias summary for randomized studies (RoB 2).

Study	Bias from Randomization Process	Bias due to Deviations from Intended Interventions	Bias due to Missing Outcome Data	Bias in Measurement of the Outcomes	Bias in Selection of the Reported Result	Overall Risk of Bias
Graven-Nielsen, 2000 [22]	Low	Low	Low	Low	Low	Low
Dailey, 2013 [23]	Low	Low	Low	Low	Low	Low
Zanette, 2014 [24]	Low	Low	Low	Low	Low	Low
Meeus, 2015 [25]	Low	Low	Low	Low	Low	Low
Pickering, 2018 [26]	Low	Low	Some concerns	Low	Low	Some concerns
Amer-Cuenca, 2019 [27]	Low	Low	Low	Low	Low	Low
Udina-Cortés, 2020 [28]	Low	Low	Low	Low	Low	Low
Ablin, 2023 [29]	Low	Some concerns	Low	Low	Low	Some concerns
Paula, 2023 [30]	Low	Low	Low	Low	Low	Low

**Table 3 biomedicines-12-02097-t003:** Association between endogenous pain modulation and clinical improvement.

Association	Number of Trial Arms	Odds Ratio	95% CI	*p*-Value
CPM improvement vs. VAS improvement	17	16	1.26 to 553.58	0.002
CPM improvement vs. FIQR improvement	14	13	1.09 to 1000.61	0.004
TS improvement vs. VAS improvement	8	6	1.58 to 426.85	0.003
TS improvement vs. FIQR improvement	6	5	1.16 to 1060.22	0.002

## Data Availability

The data that support the findings of this study are available from the corresponding author upon reasonable request.

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
