# Peer review of "The Link between Endogenous Pain Modulation Changes and Clinical Improvement in Fibromyalgia Syndrome: A Meta-Regression Analysis"

_biomedicines, 2024, doi:10.3390/biomedicines12092097_

Round 1

Reviewer 1 Report

Comments and Suggestions for Authors

The authors aimed to evaluate the association between somatosensory changes, inhibitory mechanisms, and clinical improvements under different therapeutic paradigms in patients with fibromyalgia. They achieved this by conducting a systematic review with meta-analysis using an appropriate methodology based on PRISMA guidelines. However, there are several concerns about the paper's quality.

The introduction and discussion sections of the paper do not meet the quality standards to be published in the present form. I recommend better define the outcomes of interest in the introduction section avoiding including the statement about COVID that does not help to focus the reader on the main goal of this article. I think the authors are right in identifying that there is a need to consider the dynamic features of endogenous pain modulation. However, the authors consider TS a marker in pain modulation (line 87) which is not correct. Again, I recommend to include and explicit a more precise defintion of TS and CPM to avoid this problem.

I also recommend extending the discussion section (one page only). I recommend being more critical on the results, given the high variability some paradigms (CPM). I also think authors should review the use of abbreviation since there is a clear lack of consistency across the different sections. Finally, the results section should not include any interpretation or hypothesis (Line 221-223) and the conclusion sections should be limited to what this study design allows to conclude.

I find it challenging to discern the clinical relevance of the findings regarding "a significant association between clinical improvement and the normalization of CPM and TS in fibromyalgia syndrome", given that an improvement in CPM and TS already allows to suggest a clinical improvement based on specific pain mechanisms (CPM inhibitory/TS facilitatory).

And again "The CPM and TS tests may serve as valuable surrogate biomarkers for FMS management, highlighting the potential for targeted non-pharmacological interventions to restore defective endogenous pain modulation mechanisms." This conclusion does not add anything new to the literature, and the study design is not the appropiate one to conclude this.

Comments on the Quality of English Language

There are several mistakes in the written language, including (but not limited to) "it may not be detect" Line 73, "because the different types of intervention among studies and the various ways to measure CPM." LIne 208. "Although the analysis demonstrated varied effects between the experimental and control groups." LIne 200

Author Response

Reviewer 1

  • The authors aimed to evaluate the association between somatosensory changes, inhibitory mechanisms, and clinical improvements under different therapeutic paradigms in patients with fibromyalgia. They achieved this by conducting a systematic review with meta-analysis using an appropriate methodology based on PRISMA guidelines. However, there are several concerns about the paper's quality.

Answer: Thank you for your feedback. We have addressed your comments below.

  • The introduction and discussion sections of the paper do not meet the quality standards to be published in the present form. I recommend better define the outcomes of interest in the introduction section avoiding including the statement about COVID that does not help to focus the reader on the main goal of this article. I think the authors are right in identifying that there is a need to consider the dynamic features of endogenous pain modulation. However, the authors consider TS a marker in pain modulation (line 87) which is not correct. Again, I recommend to include and explicit a more precise definition of TS and CPM to avoid this problem.

Answer: We appreciated your suggestions. We modified the introduction, accordingly, defining the outcomes better, removing the paragraph about COVID, and adding proper definitions of TS and CPM.

  • I also recommend extending the discussion section (one page only). I recommend being more critical on the results, given the high variability some paradigms (CPM).

Answer: Thank you for your comments. We have expanded the discussion, commenting the high variability of CPM paradigms

  • I also think authors should review the use of abbreviation since there is a clear lack of consistency across the different sections.

Answer: Thanks, abbreviations were corrected.

  • Finally, the results section should not include any interpretation or hypothesis (Line 221-223) and the conclusion sections should be limited to what this study design allows to conclude.

Answer: Thanks, we have corrected the Line 221-223 and the conclusion section as suggested.

  • I find it challenging to discern the clinical relevance of the findings regarding "a significant association between clinical improvement and the normalization of CPM and TS in fibromyalgia syndrome", given that an improvement in CPM and TS already allows to suggest a clinical improvement based on specific pain mechanisms (CPM inhibitory/TS facilitatory).

Answer: We appreciate your comment and the opportunity to clarify the clinical relevance of our findings regarding the significant association between clinical improvement and the normalization of Conditioned Pain Modulation (CPM) and Temporal Summation (TS) in fibromyalgia syndrome.

It is accurate that improvements in CPM and TS themselves may suggest clinical improvement, as these biomarkers reflect specific pain modulation mechanisms—CPM as an inhibitory and TS as a facilitatory mechanism. However, our research extends beyond merely correlating these biomarkers with clinical improvement. In this study, we explored how specific therapeutic interventions can influence these biomarkers and, in turn, how these changes associate with improvements in patient-reported clinical parameters such as pain intensity and symptom severity.

What adds further significance to our results is the demonstration that not only is there a correlation, but this correlation is modifiable through specific interventions, suggesting therapeutic potential for tailoring interventions based on each patient's individual biomarker response. Furthermore, our findings support the notion that these biomarkers can be used not only as state indicators but also as treatment targets, providing a more solid basis for designing personalized and more targeted therapies in fibromyalgia.

It is also important to note that the association between these biomarkers and clinical outcomes is not always as straightforward as it might seem. As highlighted in meta-analyses on the subject, such as O'Brien et al. (2018), the relationship can vary significantly, underscoring the complexity of pain mechanisms in fibromyalgia and the need for a nuanced understanding of these biomarkers in clinical contexts.

Therefore, our research emphasizes the importance of these measures not only as diagnostic tools but also as crucial components in evaluating the efficacy of interventions, paving the way for more personalized and precise medicine based on the patient's pain modulation profile.

Reference: O'Brien AT, Deitos A, Pego YT, Fregni F, Carrillo-de-la-Peña MT. Defective endogenous pain modulation in fibromyalgia: a meta-analysis of temporal summation and conditioned pain modulation paradigms. The Journal of Pain. 2018 Aug 1;19(8):819-36.

  • And again "The CPM and TS tests may serve as valuable surrogate biomarkers for FMS management, highlighting the potential for targeted non-pharmacological interventions to restore defective endogenous pain modulation mechanisms." This conclusion does not add anything new to the literature, and the study design is not the appropriate one to conclude this.

Answer: We appreciate the comment and the opportunity to discuss the contribution of our study to the existing body of literature. We understand the concern that our conclusion may seem to reiterate previously established knowledge. However, our intent was to delve deeper into how specific non-pharmacological interventions can effectively modulate biomarkers such as Conditioned Pain Modulation (CPM) and Temporal Summation (TS).

The design of our study, a systematic review and meta-analysis, was chosen purposefully to integrate and critically analyze existing data from multiple previous research studies, allowing for a more rigorous and comprehensive evaluation of the effectiveness of non-pharmacological interventions on these biomarkers. This study is particularly significant as it addresses the contradictory results from previous systematic reviews and is the first to test this "assumed" association between specific interventions and changes in CPM and TS biomarkers in a systematic manner.

Through this approach, we were able to identify and quantify the influence of these interventions on pain modulation beyond what individual studies might suggest. It is precisely this rigorous synthesis and analysis of existing data that we believe adds value to the literature, providing a more robust argument for the use of CPM and TS as useful biomarkers in personalizing treatment for patients with FMS.

We acknowledge that there is always room for improvement in study design and application, and we value suggestions that could help refine future research in this area.

  • There are several mistakes in the written language, including (but not limited to) "it may not be detect" Line 73, "because the different types of intervention among studies and the various ways to measure CPM." Line 208. "Although the analysis demonstrated varied effects between the experimental and control groups." Line 200

Answer: Thank you, we corrected these sentences

Reviewer 2 Report

Comments and Suggestions for Authors

The systemic review by Barrios et al. on “The link between endogenous pain modulation changes and clinical improvement in fibromyalgia syndrome: a meta-regression analysis” explore the association between changes in the CPM and TS tests and the clinical improvement of FMS patients who received a therapeutic intervention. The team of scientist,  compiled this review by  systematically searching for FMS randomized clinical trials with data on therapeutic interventions comparing clinical improvement (pain intensity and symptom severity reduction), CPM, and TS changes relative to control interventions. The hypothesis used behind this review was, that improvement in endogenous pain modulation will correlate with better clinical profiles in FMS patients. The review is inline the scope of journal. The compilation of data and structure of review is appropriate. However, it can be accepted after following revisions.

1.      Why only PubMed/MEDLINE and Embase databases used?

2.      Two independent reviewers (R.F.M., L.F.G.-G.), third author (K.P.-B.)…This name of authors should be in credit statement, not here. Please rewrite all such sentences.

3.      Line 175: (Ablin et al), this reference should be numbered

4.      Line 179: (Pickering et al), this reference should be numbered, carefully check the manuscript for such errors.

5.      In figure 2, what is 236 at bottom?  If possible convert it to tabular form as there are lots of data.

6.      The font size of figure 2 is too small, please increase the font and resolution.

7.      4.2. . Conclusion. Remove the full stop before conclusion.

8.      In Table 1. General characteristics of included studies, first column add reference number for better understanding.

9.      In Table 2. Risk of bias summary for randomized studies (RoB 2), why there is green and yellow colour. Is there any specific reason? 

Author Response

Reviewer 2

The systemic review by Barrios et al. on “The link between endogenous pain modulation changes and clinical improvement in fibromyalgia syndrome: a meta-regression analysis” explore the association between changes in the CPM and TS tests and the clinical improvement of FMS patients who received a therapeutic intervention. The team of scientist,  compiled this review by  systematically searching for FMS randomized clinical trials with data on therapeutic interventions comparing clinical improvement (pain intensity and symptom severity reduction), CPM, and TS changes relative to control interventions. The hypothesis used behind this review was, that improvement in endogenous pain modulation will correlate with better clinical profiles in FMS patients. The review is inline the scope of journal. The compilation of data and structure of review is appropriate. However, it can be accepted after following revisions.

Answer: Thank you for your feedback. We have addressed your comments below.

  1. Why only PubMed/MEDLINE and Embase databases used?

Answer: We appreciate your comment and recognize the importance of explaining the selection of databases used for our systematic review. For this study, we chose to include exclusively randomized controlled trials (RCTs). Considering this, we selected PubMed/MEDLINE and Embase because they are the two largest and most comprehensive databases that include this type of design. Specifically, Embase is widely recognized for its exhaustive coverage in Europe, while PubMed/MEDLINE is the primary source in the United States. Together, these databases encompass over 95% of all RCTs published today, ensuring broad and representative coverage for our review.

  1. Two independent reviewers (R.F.M., L.F.G.-G.), third author (K.P.-B.)…This name of authors should be in credit statement, not here. Please rewrite all such sentences.

Answer: We corrected this sentence.

  1. Line 175: (Ablin et al), this reference should be numbered

Answer: corrected

  1. Line 179: (Pickering et al), this reference should be numbered, carefully check the manuscript for such errors.

Answer: corrected

  1. In figure 2, what is 236 at bottom? If possible convert it to tabular form as there are lots of data.

Answer: We corrected the figure 2. We decided to maintain as a figure form instead of a tabular form to show the forest plots and the comparison among groups.

  1. The font size of figure 2 is too small, please increase the font and resolution. - Fernando

Answer: corrected

  1. 4.2. . Conclusion. Remove the full stop before conclusion.

Answer: corrected

  1. In Table 1. General characteristics of included studies, first column add reference number for better understanding. - Silvia

Answer: We do appreciate your suggestion to improve the understanding of Table 1. We added the reference number in the first column for each corresponding article.

  1. In Table 2. Risk of bias summary for randomized studies (RoB 2), why there is green and yellow color. Is there any specific reason?

Answer: We appreciate your question regarding the color coding used in Table 2 of the Risk of Bias summary for randomized studies (RoB 2). The use of green and yellow colors follows a common convention in the visual representation of bias assessment, where green indicates a "low risk of bias" and yellow represents "some concerns". This color scheme facilitates quick visual identification of the bias risk level in each assessed domain, allowing readers to efficiently understand the strengths and weaknesses of the included studies. This practice is widely used in scientific literature and systematic review reports to enhance clarity and communicate critical assessment outcomes effectively (Flemyng et al., 2023).

Reference:
Flemyng E, Moore TH, Boutron I, Higgins JP, Hróbjartsson A, Nejstgaard CH, Dwan K. Using Risk of Bias 2 to assess results from randomised controlled trials: guidance from Cochrane. BMJ Evidence-Based Medicine. 2023 Aug 1;28(4):260-6.

Reviewer 3 Report

Comments and Suggestions for Authors

The analysis explores treatment principles for fibromyalgia patients. I do agree with the methods and the results; however, I do not think that the discussion is apt. In the results the authors cannot confirm a signifiant effect pf DCM, but the discussion hinges on this effect. The discussion does not mention TS which among the results plays a major role.

Also, in „strength and limitations“ the authors claim to have reduced the risk of publication bias. However, checking the Forest plots it appears evident from the number of studies with a low effect, but nearly no studies showing a low negative effect, and the frequency of CI straddling 0 a publication bias appears to be present. A meta analysis cannot reduce publication bias, but may show the presence of this bias (or another relevant confounder not considered, but suggested by the low number of participants in the studies).

Some parts of the discussion are unclear to the average reader. Abbreviations like MPC or DCM are not explained but should, to enhance readability. The paragraph including these abbreviations appears to be out of context.

Fibromyalgia is partly a disgnosis of exclusion, based on pain in the musculosceletal system but with no other causative factor present. This suggests fibromyalgia to be clustered among the group of pain syndromes with no peripheral cause, and a strong nervous system component. This is in line with current therapies mostly including pain comedications but not classical drugs like NSAIDs or opiates.

Although the English is fine and readable, the authors should proofread their manuscript; it contains a number of gramattical errors, and faulty or incomplete sequences .

Comments on the Quality of English Language

English is fine, but the grammar needs some work.

Author Response

Reviewer 3

  • The analysis explores treatment principles for fibromyalgia patients. I do agree with the methods and the results; however, I do not think that the discussion is apt. In the results the authors cannot confirm a significant effect pf DCM, but the discussion hinges on this effect. The discussion does not mention TS which among the results plays a major role.

Answer: Thank you for your feedback. We have edited the discussion section, toned down the interpretation of the results and added the TS findings.

  • Also, in “strength and limitations” the authors claim to have reduced the risk of publication bias. However, checking the Forest plots it appears evident from the number of studies with a low effect, but nearly no studies showing a low negative effect, and the frequency of CI straddling 0 a publication bias appears to be present. A meta-analysis cannot reduce publication bias, but may show the presence of this bias (or another relevant confounder not considered, but suggested by the low number of participants in the studies).

Answer: Thank you for pointing out that. We have corrected the strength and limitations section clarifying that we did not reduce the risk of publication bias and due to the small number of included studies we could not assess its presence.

  • Some parts of the discussion are unclear to the average reader. Abbreviations like MPC or DCM are not explained but should, to enhance readability. The paragraph including these abbreviations appears to be out of context.

Answer: Thank you. We corrected the abbreviations and clarified the discussion paragraph.

  • Fibromyalgia is partly a diagnosis of exclusion, based on pain in the musculoskeletal system but with no other causative factor present. This suggests fibromyalgia to be clustered among the group of pain syndromes with no peripheral cause, and a strong nervous system component. This is in line with current therapies mostly including pain comedications but not classical drugs like NSAIDs or opiates.

Answer: Thank you, we agree with your perspective about Fibromyalgia, we interpreted our results accordingly.

  • Although the English is fine and readable, the authors should proofread their manuscript; it contains a number of grammatical errors, and faulty or incomplete sequences. English is fine, but the grammar needs some work.

Answer: Thanks, we proofread the manuscript and correct the grammatical errors.

Reviewer 4 Report

Comments and Suggestions for Authors

Dear Sir,

     I have following comments regarding your article.

Abstract: Abstract line can be modified.as overall effect was not significant.

Introduction:   

·         Anxiety, Depression, Pain, may be result of isolation and other thing in COVID underlying pathogenesis of FMS and APCS may be fallacious association.

·         Pain mechanics and modulations, is a complex process, and intricacies complexities may be elaborated.

Result: 

·         Type of Drug dose duration is case of pharmacological treatment.

·         Type of neuromodulator, frequency and dose is case of non-invasive neuromodulator.

·         RTMS not included.

·         No. of patients improved >50% or <50% (Response Rate)

·         Separate table for neuromodulator CPM/TS.

Discussion: 1st paragraph last line to be changed as pooled analysis not significant.

Authors may address other biochemical markers/study.

Conclusion: Should be based on results.  

Author Response

Reviewer 4

I have the following comments regarding your article.

  • Abstract: Abstract line can be modified.as overall effect was not significant.

Answer: Thank you, we corrected the abstract.

  • Introduction:
  • Anxiety, Depression, Pain, may be result of isolation and other thing in COVID underlying pathogenesis of FMS and APCS may be fallacious association.

Answer: Thank you for your suggestion. We deleted the COVID related paragraph because this was not helping in the main goal of this article.

  • Pain mechanics and modulations, is a complex process, and intricacies complexities may be elaborated.

Answer: Thank you for your comment, we agree with the complexity of pain perception. We clarified it in our introduction.

Result:

  • Type of Drug dose duration is case of pharmacological treatment.

Answer: We added this information in Table 1.

  • Type of neuromodulator, frequency and dose is case of non-invasive neuromodulator.

Answer: We added this information in Table 1.

  • RTMS not included.

Answer: Thank you for your feedback. We did not include the rTMS trials because none of them provide data on TS and CPM before and after the stimulation.

  • No. of patients improved >50% or <50% (Response Rate)

Answer: Thank you for your observation. We did not include this outcome because it is not an standard fibromyalgia trials. 

  • Separate table for neuromodulator CPM/TS.

Answer: Due to the small number of studies reporting CPM and TS, we decided to keep the information on one table. We believe this is beneficial for the readability and interpretability of our findings.

  • Discussion: 1st paragraph last line to be changed as pooled analysis not significant.

Answer: Thank you, we corrected the paragraph.

  • Authors may address other biochemical markers/study.

Answer: We appreciate your observation regarding the inclusion of other biochemical markers in the discussion of our study. Although fibromyalgia presents alterations in various potential biomarkers such as inflammatory markers (Andrés-Rodríguez et al., 2019), receptors like the Mu opioid receptor (Malafoglia et al., 2023), and neurotrophins such as brain-derived neurotrophic factor (BDNF) (Xiong et al., 2024; Merighi et al., 2024), we chose not to extensively focus on these aspects in the discussion due to their unstable nature and the complexity of other biomarkers, whose studies often show varied and contradictory results (Favretti et al., 2023). This variability is likely due to the influence of multiple external and internal variables that can act as confounders in this population.

In this case, we specifically focused on pain modulation markers, such as conditioned pain modulation (CPM) and temporal summation of pain (TS), due to their direct relevance in assessing therapeutic interventions for fibromyalgia and the availability of coherent and replicable data. By limiting our focus to these markers, we aimed to provide a more accurate analysis and avoid additional confusion that could arise from discussing fewer stable biomarkers. Nevertheless, we recognize the importance of future research that may include a broader range of biomarkers to better understand the underlying mechanisms in fibromyalgia, which could enrich the evidence base and improve the personalization of treatment for these patients.

  • Conclusion: Should be based on results.

Answer: Thank you, we corrected the conclusion section.